# Dynamic Metabolic Responses of Resistant and Susceptible Poplar Clones Induced by *Hyphantria cunea* Feeding

**DOI:** 10.3390/biology13090723

**Published:** 2024-09-14

**Authors:** Zheshu Wang, Liangjian Qu, Zhibin Fan, Luxuan Hou, Jianjun Hu, Lijuan Wang

**Affiliations:** 1State Key Laboratory of Tree Genetics and Breeding, Key Laboratory of Tree Breeding and Cultivation of the State Forestry Administration, Research Institute of Forestry, Chinese Academy of Forestry, Beijing 100091, China; wzs18903741937@163.com (Z.W.); fanzhibin00@163.com (Z.F.); hlxcaf@163.com (L.H.); hujj@caf.ac.cn (J.H.); 2Key Laboratory of Forest Protection of National Forestry and Grassland Administration, Ecology and Nature Conservation Institute, Chinese Academy of Forestry, Beijing 100091, China; qulj2001@caf.ac.cn; 3Co-Innovation Center for Sustainable Forestry in Southern China, Nanjing Forestry University, Nanjing 210037, China

**Keywords:** poplar, induced resistance, insects, *Hyphantria cunea*, metabolomics

## Abstract

**Simple Summary:**

Poplar forests are significantly threatened by *Hyphantria cunea* in China. Although previous studies have identified variations in resistance among the different poplar clones to this pest, the induced mechanisms behind the resistance are not yet well understood. This research investigated the dynamic changes in the defensive enzymes and metabolic profiles in resistant ‘2KEN8’ and susceptible ‘Nankang’ with or without feeding by *H. cunea*. The findings suggested that the resistant poplar clone initiated an earlier and stronger accumulation of defensive enzymes, and metabolites such as phenolic compounds, flavonoids, and unsaturated fatty acids than the susceptible one after infestation. These changes might contribute to inhibition of larva development in ‘2KEN8’. The present results are helpful for revealing the mechanisms of poplar resistance to *H. cunea* and breeding resistant varieties.

**Abstract:**

Poplar trees are significant for both economic and ecological purposes, and the fall webworm (*Hyphantria cunea* Drury) poses a major threat to their plantation in China. The preliminary resistance assessment in the previous research indicated that there were differences in resistance to the insect among these varieties, with ‘2KEN8’ being more resistant and ‘Nankang’ being more susceptible. The present study analyzed the dynamic changes in the defensive enzymes and metabolic profiles of ‘2KEN8’ and ‘Nankang’ at 24 hours post-infestation (hpi), 48 hpi, and 96 hpi. The results demonstrated that at the same time points, compared to susceptible ‘Nankang’, the leaf consumption by *H. cunea* in ‘2KEN8’ was smaller, and the larval weight gain was slower, exhibiting clear resistance to the insect. Biochemical analysis revealed that the increased activity of the defensive enzymes in ‘2KEN8’ triggered by the feeding of *H. cunea* was significantly higher than that of ‘Nankang’. Metabolomics analysis indicated that ‘2KEN8’ initiated an earlier and more intense reprogramming of the metabolic profile post-infestation. In the early stages of infestation, the differential metabolites induced in ‘2KEN8’ primarily included phenolic compounds, flavonoids, and unsaturated fatty acids, which are related to the biosynthesis pathways of phenylpropanoids, flavonoids, unsaturated fatty acids, and jasmonates. The present study is helpful for identifying the metabolic biomarkers for inductive resistance to *H. cunea* and lays a foundation for the further elucidation of the chemical resistance mechanism of poplar trees against this insect.

## 1. Introduction

Populus species, characterized by their rapid growth, are extensively utilized in ecological and economic forestry plantations across China [1]. Due to the monoculture cultivation practices and the exacerbating effects of climate change, pest infestations have become a major factor threatening the plantation of poplar trees [1,2]. *Hyphantria cunea* Drury (Lepidoptera: Erebidae), a defoliating insect native to North America that was introduced to China in 1979, poses a considerable risk to broad-leaved tree species in Northern China. Although this defoliator is polyphagous, poplar trees are among its preferred hosts [3]. In recent years, it has become a serious pest of poplar plantations in China [4,5]. Cultivating resistant varieties represents the most cost-effective, environmentally benign, and efficacious strategy for managing this pest [6]. Investigating the underlying mechanisms of tree defense against *H. cunea* will provide useful information for the breeding of resistant forest trees.

Over the long term of co-evolution with insects, plants have developed two types of defense mechanisms against herbivores—morphological and biochemical. Morphological or physical mechanisms involve the production of structures that shield the plant’s surface from insect infestation, such as hairs, trichomes, and thicker leaves [7]. Should insects overcome these morphological defenses, biochemical mechanisms are then engaged. Biochemical defense is primarily executed by metabolites that influence the feeding, growth, and survival of herbivores [8,9,10]. Primary plant metabolites, including sugars, lipids, and amino acids, determine the nutritive value of the host plant to insects and can affect insect development [11,12]. Secondary plant metabolites, which include terpenoids, alkaloids, anthocyanins, phenols, and quinones can inhibit insect growth and development, directly kill insects, or indirectly reduce herbivory by deterring insects, attracting their predators, or enticing their parasites [7,8,9,10,11,12,13,14,15,16].

Plants produce defensive compounds either constitutively or in response to insect attacks. Constitutive compounds are continuously expressed, regardless of the presence of insect attacks [17]. While constitutive resistance offers a constant protective barrier, the investment of resources into this type of resistance is inherently limited due to its higher metabolic cost [18]. In contrast, induced resistance offers a cost-effective strategy for minimizing defense-related resource expenditure. In the absence of herbivore threats, plants prioritize resource allocation for growth and reproduction over defense. Upon an herbivore attack, the plants activate the induced defense mechanisms, and a built-in trade-off between growth and resistance is established, leading to an enhanced production of defensive compounds that assist them in deterring the attackers [19,20,21].

There is a diverse array of compounds implicated in the induced resistance of plants to insects, with these compounds typically being species-specific and subject to variation throughout plant development [10]. In poplar, *Lymantria dispar* L. larvae induced the synthesis of lignin and hydrolysable tannins in the affected leaves [22,23]. In contrast to *L. dispar*, feeding by *Malacosoma disstria* resulted in an elevated level of condensed tannins in poplar [24]. The activity of aspen leaf miner larvae (*Phyllocnistis populiella*) led to increased levels of tremulacin and salicortin in aspen foliage [25]. Stem borer *Apriona germari* (Hope) induced the accumulation of quinic acid, epicatechin, epigallocatechin, and salicin in the bark, while also upregulating the levels of coniferyl alcohol, ferulic acid, and salicin in the xylem [26]. It is evident that the metabolites involved in the induced resistance of poplar trees to various insects exhibit a high degree of diversity. 

In the previous study, we assessed the resistance to *H. cunea* in seven commonly planted poplar varieties in China, identifying ‘2KEN8’ as relatively resistant and ‘Nankang’ as susceptible cultivar, respectively [27]. Hydroxycinnamates, benzenoids, and their derivatives were characterized as constitutive metabolites that contribute to poplar resistance against *H. cunea*. A subsequent small-scale experiment was conducted to elucidate the role of two specific constitutive metabolites, stigmasterol and sinapic acid, in the induced resistance of *H. cunea*-challenged poplar seedlings. The findings indicated that both metabolites were upregulated in response to *H. cunea*, but they exhibited distinct patterns post-induction between the resistant ‘2KEN8’ and the susceptible ‘Nankang’ [27]. These initial data suggested that the mechanisms of insect resistance in plants are more intricate than previously understood. In the present study, to gain a deeper understanding of the mechanisms underpinning poplar resistance to *H. cunea*, we compared the metabolic responses of the resistant ‘2KEN8’ and the susceptible cultivar ‘Nankang’ at 24 hours post-infestation (hpi), 48 hpi, and 96 hpi. The research findings revealed that ‘2KEN8’ and ‘Nankang’ employed distinct metabolic reprogramming in response to *H. cunea* feeding, leading to varying levels of resistance.

## 2. Materials and Methods

### 2.1. Plant Material and Growth Conditions

In the previous insect infestation experiments, the resistance of seven poplar varieties, including *P. deltoides* CL. ‘2KEN8’, *P. deltoides* CL. ‘55/56’, *P. deltoides* CL. ‘Nan’, *P. deltoides* CL. ‘Zhonghe-1’, *P. deltoides* CL. ‘Danhong’, *P. deltoides* CL. ‘Quanhong’ and *P. deltoides* CL. ‘Nankang’, were evaluated. Subsequently, ‘2KEN8’ and ‘Nankang’ were identified as the relatively resistant and susceptible poplar clone, respectively [27]. These two cultivars were selected as the materials for the present study. The cuttings used for propagation were from two-year-old poplar saplings with no symptoms or signs of disease and pests. These cuttings were planted in a greenhouse under controlled conditions (23 ± 1 °C; light/dark = 16:8 h). After six weeks, healthy seedlings that were approximately 50 cm in height and exhibited uniform growth were selected for infestation.

### 2.2. Insect and Rearing Conditions

The eggs of the *H. cunea* were sterilized using 10% sodium hypochlorite solution for one minute and then rinsed with sterilized water three times [28]. After the larvae hatched, they were reared with an artificial diet in plastic cups in an insectarium (23 ± 1 °C; 65% relative humidity; 16 h light/8 h dark cycle) until the fourth instar. The components and weight ratios of the artificial diet were as follows: wheat germ 7%, sucrose 4%, protein 5%, Weiser’s salts 0.8%, sorbic acid 1%, methyl 4-hydroxybenzoate 0.4%, ascorbic acid 0.17%, vitamin B 0.12%, agar 1.4%, cholesterol 0.11%, and water 80%. The larvae of *H. cunea* were provided by the Ecology and Nature Conservation Institute, Chinese Academy of Forestry. Prior to being transferred onto the poplar seedlings, the larvae were subjected to a 24 h starvation period on a piece of filter paper in a Petri dish. Larvae with similar weight (70 ± 1 mg) were selected for the infestation.

### 2.3. Treatment of Poplar Trees with H. cunea

For each cultivar, 60 seedlings were chosen for the experiment, with 30 designated for the infestation and the remaining 30 serving as the non-infestation controls. Two healthy mature leaves, the seventh and eighth leaves counting from the top, were selected. Fourth instar larvae of *H. cunea* which had begun to feed separately and had entered the voracious feeding phase were used for the infestation. Each chosen leaf was bagged with a little nylon mesh to confine the larva, and each bag contained one fourth instar larva of the *H. cunea*. The opening was secured with a string to prevent the escape of the larvae. At 24 hpi, 48 hpi, and 96 hpi, the leaf consumption was measured by Image J 1.53k software, and the weight of the larvae was recorded. For every cultivar at each time point, 10 seedlings were inoculated with *H. cunea*, and another 10 seedlings served as uninoculated controls. Among the 10 seedlings that were either inoculated or used as controls, 7 seedlings were used for metabolic profiling and 3 were used for the enzyme activity analysis. During sampling, the leaves were gently detached at the petiole to minimize metabolic responses induced by mechanical injury, and then quickly immersed in liquid nitrogen for flash-freezing, followed by storage at −80 °C for subsequent metabolite analysis.

### 2.4. Determination of the Enzymatic Activity of PPO and POD

At 24 hpi, 48 hpi, and 96 hpi, leaves from the infestation and non-infestation poplar seedlings were collected for the peroxidase (POD) and polyphenol oxidase (PPO) enzymes activity assays. The frozen leaves were taken out of the refrigerator and ground into a powder. The activities of POD and PPO enzymes in each cultivar were then determined using the POD and PPO Activity Assay Kit (Solarbio, Beijing, China), with specific assay methods detailed in the kit’s instructions [29,30]. For every cultivar at each time point, both the infestation and non-infestation samples consisted of three plants, representing three biological replicates.

### 2.5. Metabolomic Analysis of Leaf Tissues

The frozen leaf tissues were ground into a fine powder using a mortar and pestle. A sample of approximately 100 mg of this powder was used for the extraction of metabolites according to previous methods with some modifications [27,31]. The ground powder was added to 1.5 mL of the extraction solution (methanol/chloroform/water in a ratio of 5:2:2). To this mixture, the internal quantitation standards comprising 5 µL of nonadecanoic acid at a concentration of 2.0 mg/mL and 80 µL of ribitol at a concentration of 0.2 mg/mL were added. After extraction, 500 µL of the polar extract and 200 µL of the lipophilic extract were evaporated to dryness under a stream of nitrogen in a vacuum rotary evaporator, without the application of heat. To the dried residues, 50 µL of methoxy amine hydrochloride (20 mg/mL, Sigma-Aldrich, St. Louis, MO, USA) in pyridine (ACS grade) was added, and the mixture was incubated for 2 h at 30 °C to derivatize the metabolites. Following this, 100 µL of N-Methyl-N-trifluoroacetamide (MSTFA) was added to the derivatized extracts and incubated for an additional 30 min at 37 °C to complete the chemical reaction. The derivatized samples were then analyzed using a LECO Pegasus IV GC-TOF/MS system equipped with a DB-5 ms capillary column (0.25 mm, 30 m × 0.32 mm) for the separation of the metabolites. The temperature program for the chromatographic separation of polar and non-polar compounds was based on the method previously described by Wang et al. [27].

### 2.6. Data Processing and Statistical Analyses

GC-TOF/MS data were acquired and processed with Chroma-TOF software version 4.0 (LECO). The data underwent mean-centering and autoscaling to achieve a unit variance, followed by analysis using partial least squares regression-discriminant analysis (PLS-DA) with the SIMCA-P 12.0 software package (Umetrics, Umeå, Sweden). In the PLS-DA model’s loadings plot, the metabolites identified by a variable importance for the project (VIP) > 1.0 were considered responsible for profile differences and were selected for further examination. Then, nonparametric Mann-Whitney U tests were conducted to compare the abundance of the above metabolites across different cultivars. Metabolites were deemed differential if they met the following two criteria: a VIP score greater than 1.0 and a Mann-Whitney *p*-value below 0.05. These differential metabolites were identified by comparing the mass of the fragments with the National Institute of Standards and Technology’s (NIST, USA) standard mass spectral databases using a similarity of 80%. Quantitative enrichment analysis of the differential metabolites was carried out in the web-based Metabo-Analyst 6.0 software based on the pathway library of *Arabidopsis thaliana*. Statistical analysis was performed by SPSS version 17.0 (SPSS Inc., Chicago, IL, USA). Student’s *t*-test was used to compare the differences in average weight between larvae feeding on the two cultivars (*p* < 0.05). One-way analysis of variance (ANOVA) and Duncan’s multiple range tests were conducted to determine significant difference among the inoculated and control samples (*p* < 0.05).

## 3. Results

### 3.1. Quantification of Leaf Damage in Resistant and Susceptible Poplar Clone

The seedlings were infested with the larvae of *H. cunea* as shown in Figure 1A. Following infestation, the extent of the leaf area consumed and the corresponding larval weight were documented to ascertain the comparative resistance levels of the two cultivars. After feeding for the same duration, the consumed leaf area of ‘2KEN8’ was significantly smaller than that of ‘Nankang’. Figure 1B shows the leaves of the two cultivars at 48 hpi. The diminished feeding area on the ‘2KEN8’ leaves suggests a potential for reduced weight gain in the larvae that feed on this cultivar compared to those feeding on the ‘Nankang’ leaves. Indeed, the measurements of larval average weight indicate that although the larvae’s weight was similar before infestation for both cultivars, after 96 h of feeding, there was a significant change in weight between larvae feeding on the two varieties. The average weight of the larvae feeding on the ‘Nankang’ leaves was 220 mg, with an increase of more than twice the body weight, while the larvae feeding on the ‘2KEN8’ leaves only had an average weight of 140 mg, with an increase of only one-fold in the body weight (Figure 1C). These phenotypic outcomes corroborated that ‘2KEN8’ was more resistant to *H. cunea* compared to ‘Nankang’.

### 3.2. Changes in the Activity of Defensive Enzymes

The activities of POD and PPO, which are pivotal enzymes in the response to biotic stress, were examined. Initial findings revealed no discernible disparities in the enzymatic activities of POD and PPO between the two cultivars under non-infestation conditions. Following infestation, a clear induction of POD activity, approximately 2- to 3-fold, was observed in both cultivars. Specifically, at 24 hpi and 48 hpi, the POD activity in ‘2KEN8’ was markedly elevated compared to that of ‘Nankang’ (Figure 2A). Paralleling the modulation observed in POD, the activity of PPO was also upregulated in response to the feeding by *H. cunea* with an increase ranging from 1.2 to 3 times. Notably, the PPO activity in the infested ‘2KEN8’ was approximately 1.7-fold that of ‘Nankang’ (Figure 2B).

### 3.3. Dynamic Metabolic Responses in Resistant and Susceptible Cultivar Induced by H. cunea

The dynamic metabolic profiles of ‘2KEN8’ and ‘Nankang’, both infested and non-infested, were analyzed. The findings revealed distinct trajectories in the metabolic profiles of ‘2KEN8’ and ‘Nankang’ following the infestation with *H. cunea*. At 24 hpi, a pronounced divergence was observed between the infestation and control samples of ‘2KEN8’, indicating a robust induction of metabolic profile alterations by *H. cunea*. By 48 hpi, the changes induced by *H. cunea* were further intensified. At 96 hpi, the differences between the two groups diminished (Figure 3). Contrary to the trajectory observed in ‘2KEN8’, at 24 hpi, no significant separation was detected between the infested and control samples of ‘Nankang’, suggesting minimal metabolic changes post-infestation. However, by 48 hpi, the separation between the treated and control groups of ‘Nankang’ emerged, which became more pronounced at 96 hpi (Figure 3). These results suggested that, in comparison to the susceptible ‘Nankang’, the resistant ‘2KEN8’ initiated and exhibited an earlier and stronger metabolic profile response to *H. cunea*.

### 3.4. Differential Metabolites Induced by H. cunea in ‘2KEN8’ and ‘Nankang’

Differential metabolites induced by the feeding of *H. cunea* at the three time points in ‘2KEN8’ and ‘Nankang’ were identified. Compared to their respective non-infestation controls, at 24 hpi, the feeding of *H. cunea* induced 80 differential metabolites in ‘2KEN8’ and 35 in ‘Nankang’, with 23 of these being common differential metabolites (Figure 4A). At 48 hpi, the number of differentially expressed metabolites increased to 111 in ‘2KEN8’ and 94 in ‘Nankang’, with 52 metabolites being shared between the two cultivars (Figure 4B). At 96 hpi, the induction of differential metabolites reached 102 in ‘2KEN8’ and 128 in ‘Nankang’, with 60 metabolites being common to both (Figure 4C). Among these, 32 differential metabolites were consistently identified across all three time points in the resistant ‘2KEN8’, while in the susceptible ‘Nankang’, 15 differential metabolites were consistently observed (Figure 4D,E).

### 3.5. Pathways Associated with Resistance to H. cunea in Poplar

Metabolomic analysis indicated that the differential metabolites induced by *H. cunea* are significantly enriched in the following pathways, primarily including the phenylpropanoid pathway, flavonoid biosynthesis, and biosynthesis of unsaturated fatty acids. Within the phenylpropanoid pathway, phenylalanine, cinnamic acid, coumaric acid, and sinapic acid exhibited constitutive differences between the two cultivars. The content of phenylalanine in the resistant ‘2KEN8’ was significantly lower than that of the susceptible ‘Nankang’. However, the contents of cinnamic acid, coumaric acid, and sinapic acid were higher in ‘2KEN8’ than in ‘Nankang’. These compounds were also induced by *H. cunea* infestation. After infestation, the level of phenylalanine was induced to decrease at all three time points. Contrary to the change in phenylalanine, the contents of cinnamic acid and coumaric acid were induced to increase at 48 hpi and 96 hpi, and that of sinapic acid was elevated at 24 hpi and 48 hpi. For cinnamic acid, coumaric acid, and sinapic acid, the fold increase in the resistant ‘2KEN8’ was slightly higher than susceptible ‘Nankang’ following infestation (Figure 5).

In the flavonoid biosynthesis pathway, there were no significant differences in the levels of naringenin, luteolin, tricetin, and epigallocatechin between the non-infested ‘2KEN8’ and ‘Nankang’. However, these compounds were significantly induced upon feeding of *H. cunea*. Interestingly, at 24 hpi, the fold increase in these compounds within ‘2KEN8’ was markedly higher than in the susceptible ‘Nankang’, with the levels in ‘2KEN8’ being 1.5 to 3 times higher than those in ‘Nankang’ after infestation (Figure 5). Similarly, in the pathway of biosynthesis of unsaturated fatty acids, hexadecanoic acid, octadecanoic acid, oleic acid, linoleic acid, and α-linolenic acid did not show significant differences between the non-infested ‘2KEN8’ and ‘Nankang’, but their amounts were upregulated after infestation. Unlike changes in the flavonoids, in the resistant cultivar ‘2KEN8’, these compounds were significantly upregulated at 24 hpi and continued to be increased at 48 hpi and 96 hpi, while in the susceptible cultivar ‘Nankang’, there was no significant change in the above compounds at 24 hpi, with some compounds being induced at 48 hpi and a broad induction occurring only at 96 hpi (Figure 6).

## 4. Discussion

Chemical resistance is a pivotal defense mechanism for plants to fend off pests. In this process, the metabolic products of plants, encompassing both primary and secondary metabolites, play an essential role. Variations in the levels of primary metabolites can restrict the feeding behavior of pests, while secondary metabolites exert their pest-resistant effects by causing mortality or inhibiting the growth and development of the pests [16]. The present study analyzed the dynamic metabolic profiling of the resistant ‘2KEN8’ and susceptible ‘Nankang’ following feeding by *H. cunea*. The findings revealed significant differences in the metabolic trajectories between the two cultivars. Specifically, the resistant cultivar exhibited an earlier initiation and a greater magnitude of metabolic changes. Further analysis indicated that the significantly altered metabolites primarily include phenolics, flavonoids, and unsaturated fatty acids. These substances are closely associated with the phenylpropanoid pathway, flavonoid biosynthesis, and the biosynthesis of unsaturated fatty acids.

### 4.1. The Resistant ‘2KEN8’ Exhibited Higher Levels of Defensive Enzymes after Infestation with H. cunea

Upon herbivore feeding, plants typically induce a burst of reactive oxygen species (ROS) which has various detrimental effects on plants, including increasing the permeability of cell membranes, damaging cell structure and function, and inhibiting plant growth [32]. The defensive enzyme POD can effectively scavenge reactive oxygen species, protecting plants from ROS-induced damage and aiding in the defense against insect pests. PPO is a class of copper-containing oxidoreductases that catalyze the formation of quinones from phenolic compounds, thereby reducing the nutritional value of plants and deterring insect feeding [33]. In our study, it was found that there was no significant difference in the activity of PPO and POD in the leaves of the two varieties before infestation. After insect infestation, both PPO and POD were upregulated, but the upregulation was more pronounced in the resistant ‘2KEN8’ compared to the susceptible ‘Nankang’. These results suggested that ‘2KEN8’ possessed a more robust defensive oxidase enzyme system to restrict feeding by *H. cunea*.

### 4.2. Feeding of H. cunea Triggered Stronger Metabolic Responses in the Resistant ‘2KEN8’

Induced resistance is a cost-effective and efficacious strategy commonly employed by plants to defend against insect herbivory. Metabolites play a significant role in insect-feeding-induced resistance responses [16]. Research on various plant-insect interactions has shown that the common metabolites involved in the induced responses mainly include phenolic acids, tannins, flavonoids, alkaloids, terpenoids, steroids, fatty acid derivatives, and glycosides [16,17,18]. These defensive metabolites are often species-specific. In poplar trees, hydrolysable tannins have been reported to participate in induced resistance against *Lymantria dispar*, phenolic glycosides in resistance responses to *Phyllocnistis populiella*, and phenolic acids and phenolic glycosides in resistance reactions to *Apriona germari* (Hope) [22,25,26]. It is likely that even within the same species, the resistance compounds elicited in response to different insects’ feeding may differ. The present study found that feeding by *H. cunea* induced accumulations of phenolic acids, flavonoids, and unsaturated fatty acids in poplar leaf tissues. In the resistant ‘2KEN8’, the accumulations occurred earlier and were more substantial, suggesting that the three kinds of metabolites are involved in the poplar’s induced resistance response to *H. cunea*.

### 4.3. The Phenylpropanoid Pathway Is Associated with Induced Poplar Resistance to H. cunea

The pathway of phenylpropanoid biosynthesis, which begins with the metabolism of phenylalanine, culminates in the synthesis of phenolic compounds. These compounds are integral to the formation of lignin, a critical component of plant cell walls. Extensive research has established phenolic substances as a pivotal class of anti-herbivory compounds as reviewed by Lattanzino et al. [14] and Mithöfer et al. [16]. They function either as constitutive or inducible defenses, primarily by inhibiting insect feeding or diminishing the efficacy of insect digestive enzymes. In poplar trees, phenylpropanoid compounds were recognized as the primary constituents of constitutive resistance against *H. cunea* [27]. Additionally, phenolic compounds have been implicated in the poplar’s resistance response to *Anoplophora chinensis* infestations [26]. Studies on other tree species, such as the Manchurian ash, have revealed that hydroxycoumarins and phenolic acids, including caffeic, ferulic, and *p*-coumaric acids, were instrumental in defending against the emerald ash borer (EAB) [34,35]. In oak (*Quercus rubra* L.), elevated levels of hydrolyzable tannins and polyphenolics were thought to enhance resistance to *Lymantria dispar*, potentially by inducing nutritional stress [36]. Similarly, the resistance of European oak to *Tortrix viridana* had been linked to high levels of polyphenolic compounds [37]. This current study has identified that, within the pathway, the concentrations of cinnamic acid, coumaric acid, and sinapic acid in the non-infested ‘2KEN8’ poplar were markedly higher than those in the non-infested ‘Nankang’. This disparity suggested a role for these compounds in the constitutive resistance of poplar trees to *H. cunea*. Furthermore, feeding by *H. cunea* triggered an increase in the levels of these metabolites, with the resistant ‘2KEN8’ showing a significantly greater increase than the susceptible ‘Nankang’, underscoring the involvement of these phenolic compounds in induced resistance to *H. cunea*.

### 4.4. Flavonoid and Unsaturated Fatty Acids Contribute to the Earlier Defensive Responses in ‘2KEN8’

Flavonoids play an important role in the interaction between plants and insects. Beyond their antioxidant capabilities that shield plant cells from oxidative stress, these bioactive compounds are known to curb insect feeding and oviposition behaviors, impede insect developmental processes and, in some cases, induce lethal effects on insects [38,39]. In *Pinus banksiana*, specific flavonoids such as rutin and quercetin-3-glucoside demonstrated inhibitory effects on the development of *Lymantria dispar*, escalating their mortality rates [40]. Research on peanuts indicated a correlation between the heightened levels of quercetin and rutin glycosides and the increased mortality in the tobacco armyworm, *Spodoptera litura* [41]. Moreover, in tomato (*Solanum lycopersicum*) and carrot (*Daucus carota* L.), a relationship between luteolin content and resistance to thrips was identified [42,43]. In tea plants, catechin, epicatechin, and epigallocatechin gallate have been recognized as significant inducible defensive compounds against *Ectropis grisescens* [44]. In the present study, during the early stage of infestation, *H. cunea* feeding triggered an enhanced production of epigallocatechin, tricetin, luteolin, and naringenin in the resistant cultivar ‘2KEN8’ as opposed to the susceptible ‘Nankang’. Given the diminished feeding area on ‘2KEN8’ leaves and the lesser gain of larval body weight observed with this cultivar, it was postulated that these flavonoids might play a role in deterring larval feeding.

In the present study, the levels of several metabolites in the pathway for biosynthesis of unsaturated fatty acids, including α-linolenic acid, linoleic acid, and oleic acid, were significantly increased after feeding by *H. cunea* and were further enhanced in the resistant ‘2KEN8’. These findings were consistent with the previous studies which identified that an increase in the levels of unsaturated fatty acids were associated with the resistance to insects [27,45]. The unsaturated fatty acid, α-linolenic acid, is a precursor for the synthesis of jasmonic acid (JA), which is a crucial signaling molecule in plant defense responses to insects [46,47,48]. Changes in the levels of α-linolenic acid indicated that the JA-mediated defense response was initiated earlier in the resistant ‘2KEN8’ than in the susceptible ‘Nankang’. Additionally, some studies have shown that the signaling molecule JA can regulate the accumulation of flavonoids. Exogenous treatment with methyl jasmonate (MeJA) could enhance the content of flavonoids in plants, thereby improving their resistance to pests [49,50,51,52]. So, it was likely that feeding by *H. cunea* induced an intensification of unsaturated fatty acid synthesis, leading to a rapid accumulation of the JA precursors. This could result in the accumulation of JA and the activation of the JA signaling pathway, thereby promoting the early accumulation of flavonoids and contributing to the inhibition of feeding and development of *H. cunea* larvae.

## 5. Conclusions

The present study identified that larvae feeding on the ‘2KEN8’ showed a significantly slower weight gain and reduced leaf consumption compared to those on the ‘Nankang’, thereby confirming that ‘2KEN8’ possesses greater resistance to *H. cunea* than ‘Nankang’. Analysis on the dynamic metabolic responses of the two cultivars to the insect showed that the resistant ‘2KEN8’ initiated earlier and stronger defense responses, including a higher increase in the activity of the defensive enzymes and an increased accumulation of compounds such as phenolics, flavonoids, and unsaturated fatty acids. Higher levels of the defensive enzymes and metabolites might restrict the feeding of larvae, thereby contributing to the resistance of ‘2KEN8’ to *H. cunea*. The present study provides an investigation on chemical defense of trees to *H. cunea* and lays a foundation for further elucidation of the molecular mechanisms underlying tree resistance to this insect.

## Figures and Tables

**Figure 1 biology-13-00723-f001:**
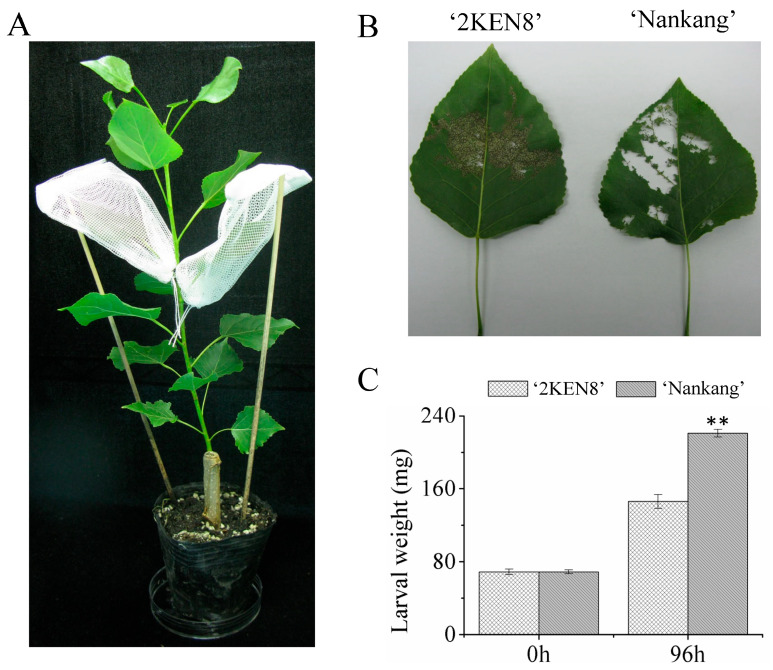
Evaluation of resistance to *H. cunea* in the two poplar clones. (**A**) Two newly mature leaves, the seventh and eighth leaves of the seedling were used for larval infestation; (**B**) Leaf consumption at 48 hpi; (**C**) Changes of larva average weight when feeding for 0 h and 96 h. The two stars indicated a significant level with a *p*-value less than 0.01.

**Figure 2 biology-13-00723-f002:**
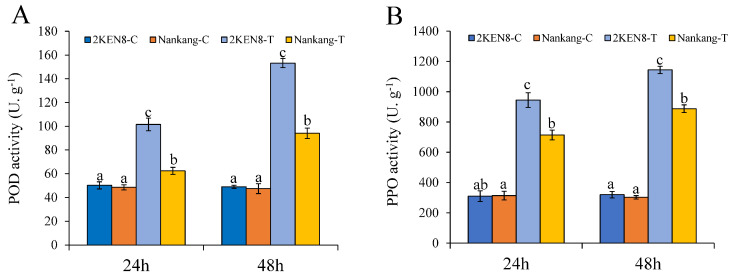
Changes in POD and PPO activities in leaves of the infested and the control group at 24 hpi and 48 hpi. (**A**) POD activity; (**B**) PPO activity. C: control group. T: infested group. Duncan’s multiple range tests were performed to determine significant difference among inoculated and control samples. Different letters in the figure indicated significant differences (*p* < 0.05).

**Figure 3 biology-13-00723-f003:**
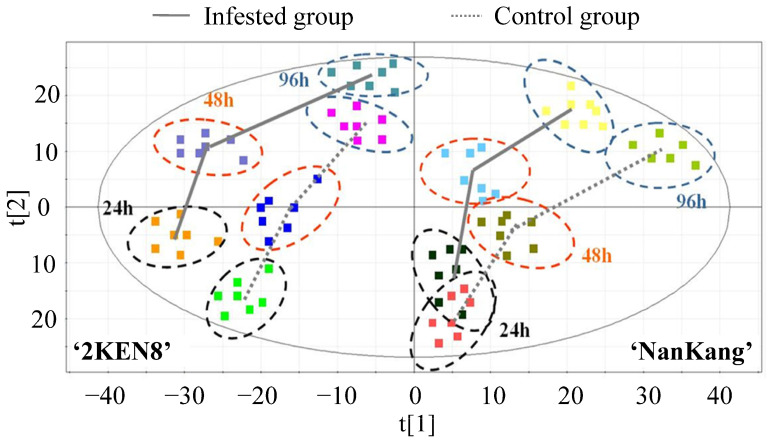
PLS-DA score plot of the infested and control samples at 24 hpi, 48 hpi, and 96 hpi. The ellipses represented the Hotelling T2 with 95% confidence. t [1] and t [2] were the first and second principal component, respectively. Each square represented an individual sample. The squares with same color were 7 replicates of each material at the same time point in infested or control group. The samples on the left side of the figure were the control and infested groups of the resistant ‘2KEN8’, while those on the right side were the control and inoculated samples of the susceptible ‘Nankang’. The solid lines represented the trajectories of the inoculated samples, while the dashed lines represented the trajectories of the control samples.

**Figure 4 biology-13-00723-f004:**
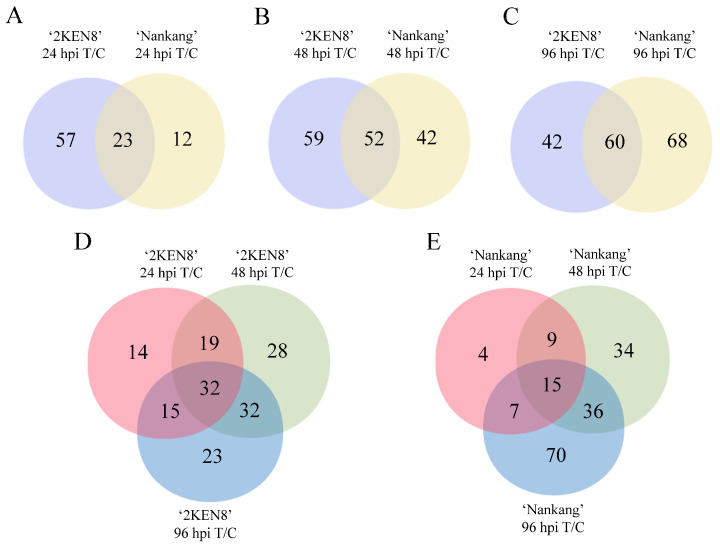
Number of differential metabolites between control and infested samples. (**A**–**C**) Comparison of the differential metabolites induced by feeding of *H. cunea* in ‘2KEN8’ and ‘Nankang’ at 24 hpi (**A**), 48 hpi (**B**), and 96 hpi (**C**), respectively. The light purple and pale-yellow circles represented the differential metabolites between the infested (T) and control (C) group for ‘2KEN8’ and ‘Nankang’, respectively. (**D**,**E**) Number of differential metabolites induced by *H. cunea* at the three time points in ‘2KEN8’ (**D**) and ‘Nankang’ (**E**). The pink, pistachio, and sky-blue circles represented the number of differential metabolites between the infested and control samples at 24 h, 48 h, and 96 h, respectively. T/C: Differential metabolites between infested and control group.

**Figure 5 biology-13-00723-f005:**
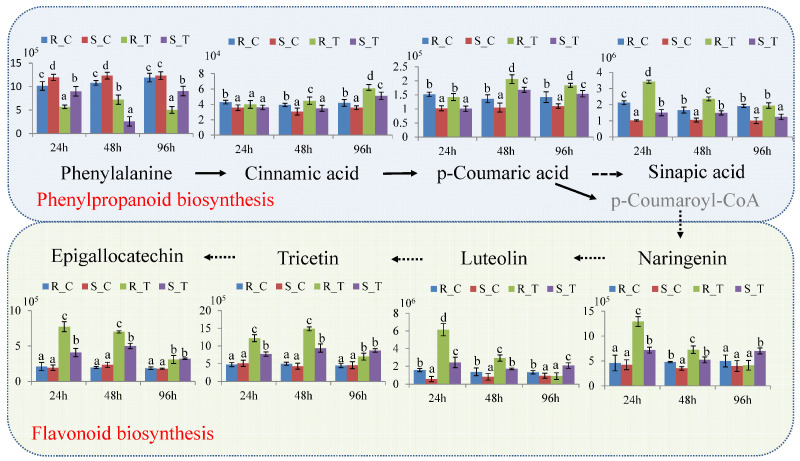
Relative contents of differential metabolites in pathways of phenylpropanoid and flavonoid biosynthesis. R: resistant ‘2KEN8’. S: susceptible ‘Nankang’. C: control group. T: infested group. Duncan’s multiple range tests were performed to determine significant difference among infested and control samples. Different letters in the figure indicated significant differences (*p* < 0.05).

**Figure 6 biology-13-00723-f006:**
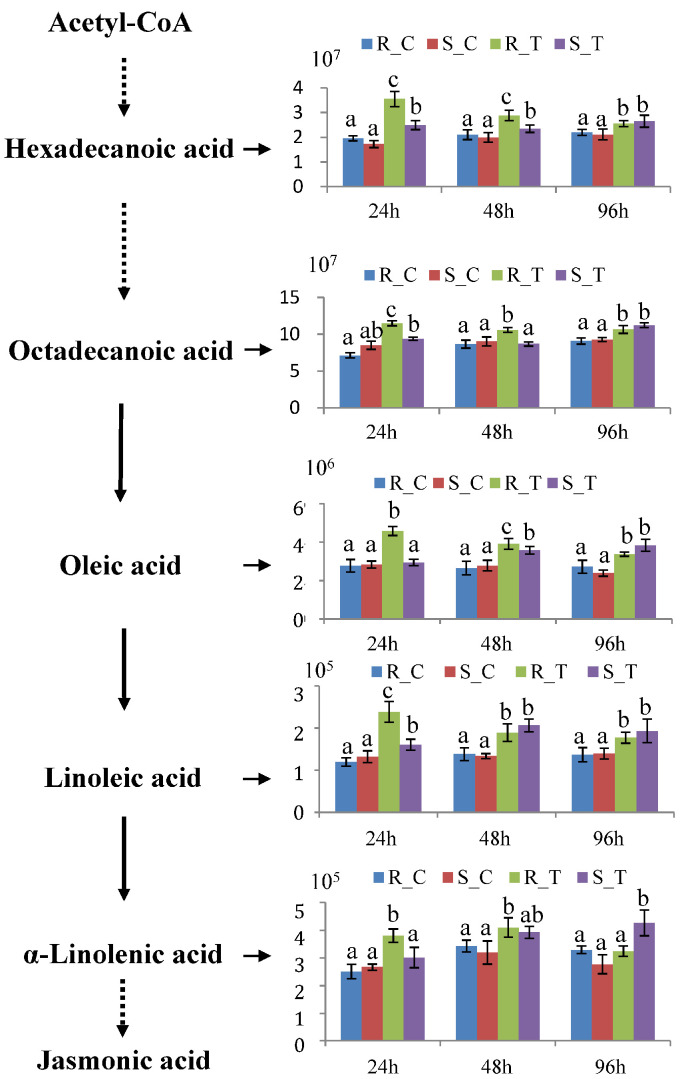
Relative abundance of differential metabolites in the pathway for biosynthesis of unsaturated fatty acids. R: resistant ‘2KEN8’. S: susceptible ‘Nankang’. C: control group. T: infested group. Duncan’s multiple range tests were performed to determine significant difference among infested and control samples. Different letters in the figure indicated significant differences (*p* < 0.05).

## Data Availability

The data presented in this study are available on request from the corresponding author.

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
