# Peer review of "Dynamic Metabolic Responses of Resistant and Susceptible Poplar Clones Induced by Hyphantria cunea Feeding"

_biology, 2024, doi:10.3390/biology13090723_

Round 1
Reviewer 1 Report
Comments and Suggestions for Authors
Dear authors,
All comments and correction suggestions are in the enclosed report.
Best wishes.

Comments on the Quality of English LanguageThe most critical errors are displayed in the report. All in all, the English level is more than satisfactory.
Author Response
Dear editor:
Thank you very much for your attention and careful consideration to our manuscript. We appreciate you very much for your positive and constructive comments and suggestions on our manuscript.
We have carefully considered your suggestion and comments and revised our manuscript according to these precious comments. We have submitted revision manuscript using online system, and we also have uploaded a marked-up copy of the changes made from the previous manuscript as attachment by the E-mail.
The main corrections and the responds to the comments point to point are listed blow:
Responses to the comments and suggestions of Reviewer #1
The subject of the study is an interesting one. However, it needs some improvements to reach publication level, as it seriously lacks details in methodology and clarity in results. The paragraphs are very uneven, sometimes vague, sometimes extremely precise and well referenced. Failing to mention the species of the study is unacceptable. A lot of references are missing. No statistical tests are clearly shown.
Reply: Thank you for your suggestion. We added the detail information for the methodology and rewrote some confusing parts in the results to make a clear presentation. We also supplemented some references to support the description in the manuscript. In the Material and Methods, we added the information about the species and cultivars used in the previous and present studies, and supplemented the statistical tests for the data analysis. We hope the revision meet with approval.
The following addresses some points as improvement suggestions.
Simple summary
L.22-23: The present results will be are helpful for revealing the mechanisms of poplar resistance to H. cunea and breeding for resistant varieties.
Reply: Thank you for your correction. We corrected the sentence in the revised version.
Page 1-line L22-23 in the revised version: “The present results are helpful for revealing the mechanisms of poplar 22 resistance to H. cunea and breeding resistant varieties.”
Abstract
L.26: the seven clones reference is useless here, leave it for the intro. “previous research” is enough.
Reply: We revised this sentence in the return version.
Page 1-line L25-26 in the revised version: “The preliminary resistance assessment in the previous research indicated that there were differences in resistance to the insect among different varieties, with ‘2KEN8’ being more resistant and ‘Nankang’ being more susceptible.”
L.28 (and all along the manuscript): “this study” could be refereeing to the previous citation. Consider changing for “our study” or “the presented study”.
Reply: We conducted a thorough check throughout the manuscript and replaced “this study” with “the presented study”.
L.32: Biochemical analysis revealed that the increase in the activity
Reply: Thank you for your suggestion. We revised this sentence in the return version.
Page 1-line L32 in the revised version: “Biochemical analysis revealed that the increased activity of defensive….”
L.37: which were are related to the biosynthesis pathways of phenylpropanoids
Reply: We revised this sentence in the revised manuscript.
Page 1-line L37 in the revised version: “which are related to the biosynthesis pathways of phenylpropanoids…..”
L.39-40: provides potential candidate markers for the assisted selection of resistant poplar trees. That’s incorrect. Unless otherwise indicated, “markers” followed by “selection” refer to genetics, which is absolutely not the subject here. Rephrase.
Reply: Thank you for your suggestion. We rewrote this sentence in the revised manuscript.
Page 1-line L38-40 in the revised version: “The present study is helpful for identifying metabolic biomarkers for inductive resistance to the H. cunea, and lays a foundation for further elucidating the chemical resistance mechanism of poplar trees against this insect.”
- Introduction
- the first part of the introduction lacks references
L.45: ref about the poplars in China
Reply: We added the reference in the revised manuscript.
Page 2-line L65 in the revised version: “…economic forestry plantations across China [1]”
[1] Hu, J.J.; Wang, L.J; Yan, D.H.; Lu, M.Z. Research and application of transgenic poplar in China. Challenges and Opportunities forthe World's Forests in the 21st Century. 2014, 567-584. http://doi.org/10.1007/978-94-007-7076-8_24.
L.46: ref about the pest infestation in poplar trees
Reply: We added the references in the revised manuscript.
Page 2-line L67 in the revised version: “…a major factor threatening the plantation of poplar trees [1, 2]”
[1] Hu, J.J.; Wang, L.J; Yan, D.H.; Lu, M.Z. Research and application of transgenic poplar in China. Challenges and Opportunities forthe World's Forests in the 21st Century. 2014, 567-584. http://doi.org/10.1007/978-94-007-7076-8_24.
[2] Liu, B.; Yan, J.Y.; Wang, D.; Wang, Y.; Zhou, Y.T.; Chen, Y.F. Occurrence of major forest pests in China in 2023 and prediction for trend in 2024. Forest pest and disease. 2024, 43, 1, 41-45. (In Chinese)
L.47: ref about H. cunea. A bit more details about the species: generalist with a preference for poplar? Specialist of Populus? When has it been introduced to China?
Reply: We added the above information and rewrote the sentence in the revised manuscript.
Page 2-line L67-70 in the revised version: “Hyphantria cunea Drury (Lepidoptera: Erebidae), a defoliating insect native to North America that has been introduced to China in 1979, poses a considerable risk to broad-leaved tree species in northern China. Although this defoliator is polyphagous, poplar trees are among its preferred hosts [3]. In recent years, it has become a serious pest of poplar plantations in China [4,5].”
[3] Ning, J.; Lu, P.; Fan, J.; Ren, L.; Zhao, L. American fall webworm in China: A new case of global biological invasions. The Innovation. 2022, 3, 1, 100201. https://doi.org/10.1016/j.xinn.2021.100201.
[4] Jiang, J.; Fan, G.; Wang, R.; Yao, W.; Zhou, B.; Jiang, T. Multi-omics analysis of Populus simonii × P. nigra leaves under Hyphantria cunea stress. Front. Plant Sci. 2024, 15, 1392433. https://doi.org/ 10.3389/fpls.2024.1392433.
[5] Ding, Y.; Shen, J.; Li, H.; Sun, Y.; Jiang, T.; Kong, X.; Han, R.; Zhao, C.; Zhang, X.; Zhao, X. Physiological and molecular mechanism of Populus pseudo-cathayana × Populus deltoides response to Hyphantria cunea. Pestic. Biochem. Phys. 2024, 202, 105969. https://doi.org/10.1016/j.pestbp.20 24.105969.
L.52: ref about the benefits of selection
Reply: We added the reference in the revised manuscript.
Page 2-line L73 in the revised version: “Cultivating resistant varieties represents the most cost-effective, environmentally benign, and efficacious strategy for managing this pest [6]”
[6] Liu, F.; Li, Q. Occurrence, Forest control status and prospect of fall-webworm (Hyphantria cunea Drury) in China. Journal of Shenyang Agricultural University. 2022, 53, 5, 630-640. https://doi.org/ 10.3969/j.issn.1000-1700.2022.05.013. (In Chinese)
L.54-55: the identification of candidate genes and biomarkers, thereby expediting the development of resistant forest trees. Rephrase. It’s not a genetics paper and this study does not address this topic. It can be mentioned but without making it sound like it’s one of the aims of the study.
Reply: Thank you for your suggestion. We corrected this sentence in the revised manuscript.
Page 2-line L73-74 in the revised version: “Investigating the underlying mechanisms of tree defense against H. cunea will provide useful information for breeding of resistant forest trees.”
L.75-77: does the triggered defense also has a trade-off cost?
Reply: We rewrote this sentence in the revised manuscript to make an easy understanding.
Page 2-line L90-97 in the revised version: “While constitutive resistance offers a constant protective barrier, the investment of resources into this type of resistance is inherently limited due to its higher metabolic cost [18]. In contrast, induced resistance offers a cost-effective strategy for minimizing the defense-related resource expenditure. In the absence of herbivore threats, plants prioritize resource allocation for growth and reproduction over defense. Upon herbivore attack, the plants activate induced defense mechanisms, and a built-in trade-off between growth and resistance is established, leading to an enhanced production of defensive compounds that assist them in deterring the attackers [19-21].”
L.90-92: which poplar? Which species?
Reply: We added the above information in 2.1. Plant Material and Growth Conditions of the revised manuscript.
Page 3-line L196-199 in the revised version: “In previous insect infested experiments, resistance of seven poplar varieties, including P. deltoides CL. ‘2KEN8’, P. deltoides CL. ‘55/56’, P. deltoides CL. ‘Nan’, P. deltoides CL. ‘Zhonghe-1’, P. deltoides CL. ‘Danhong’, P. deltoides CL. ‘Quanhong’ and P. deltoides CL. ‘Nankang’, were evaluated. ‘2KEN8’ and ‘Nankang’ were identified as the relatively resistant and susceptible poplar clone, respectively.”
L.94-96: was this subsequent small-scale experiment published?
Reply: This small-scale experiment has been published. We added the literature in the revised manuscript. In this small-scale experiment, we only investigated the changes in the content of two constitutive compounds, stigmasterol and sinapic acid, after feeding by the H. cunea. It did not mention the changes in the whole metabolic profile.
Page 3-line L200 in the revised version: “…. but they exhibited distinct patterns post-induction between the resistant ‘2KEN8’ and the susceptible ‘Nankang’ [27]”
[27] Wang, L.J.; Qu, L.J.; Hu, J.J.; Zhang, L.W.; Tang, F.; Lu, M.Z. Metabolomics reveals constitutive metabolites that contribute resistance to fall webworm (Hyphantria cunea) in Populus deltoides. Environ Exp Bot. 2017, 136, 31-40. https://doi.org/10.1016/j.envexpbot.2017.01.002.
L.102-104: reason behind these time frames?
Reply: Before conducting the present experiment, we performed a preliminary analysis on the control and treated groups of resistant cultivars ‘2KEN8’ at 12h, 24h, 48h, 72h, and 96h after infestation with H. cunea. The results showed that the metabolic profiles between 12hpi and 24hpi, as well as 48hpi and 72hpi, were relatively similar. Therefore, we chose three time points (24h, 48h, and 96h) with more pronounced changes in the metabolic profile, to analyze the difference between resistant and susceptible varieties.
2.Material and Methods
L.108-109: more details are needed! Which species, which varieties? Which studies? Only the two mentioned before or more?
Reply: We added the above information in the revised manuscript.
Page 3-line L196-199 in the revised version: “In previous insect infested experiments, resistance of seven poplar varieties, including P. deltoides CL. ‘2KEN8’, P. deltoides CL. ‘55/56’, P. deltoides CL. ‘Nan’, P. deltoides CL. ‘Zhonghe-1’, P. deltoides CL. ‘Danhong’, P. deltoides CL. ‘Quanhong’ and P. deltoides CL. ‘Nankang’, were evaluated. ‘2KEN8’ and ‘Nankang’ were identified as the relatively resistant and susceptible poplar clone, respectively [27].”
L.111-112: “free from diseases and pest”: how was it checked? Only visually? Diseases can be dormant.
Reply: Thank you for your suggestion. We rewrote this sentence in the revised manuscript. Page 3-line L201-202 in the revised version: “The cuttings used for propagation were from two-years-old poplar saplings with no symptoms or signs of disease and pests”.
L.118: Eggs of the H. cunea. Why were they sterilized? It’s not regular protocol so it needs justification.
Reply: We apologize for not providing a clear description. The sterilization referred to here is the disinfection of the egg surface, which is a common operation in culture of H. cunea in the laboratory. The purpose is to kill bacteria or viruses on the egg surface, thereby increasing the hatching rate of the eggs and the survival rate of the larvae. We corrected the sentence and added the literature in the revised manuscript.
Page 3-line L207-208 in the revised version: “Eggs of the H. cunea were sterilized using 10% sodium hypochlorite solution for one minute and then rinsed with sterilized water three times [28].”
[28] Li, S.Y.; Yu, X.H.; Fan, B.Q.; Hao, D.J. A gut‐isolated enterococcus strain (HcM7) triggers the expression of antimicrobial peptides that aid resistance to nucleopolyhedroviral infection of Hyphantria cunea larvae. Pest Manag. Sci. 2023, 79, 10, 3529-3537. https://doi.org/10.1002/ps.7533.
L.124: no discernible difference in weight: how was it tested? What’s in confidence interval?
Reply: Thank you for your suggestion. We rewrote this sentence in the revised manuscript.
Page 3-line L216-217 in the revised version: “Larvae with similar weight (70 ± 1mg) were selected for infestation.”
L.126: an indication of the phenological stage would be clearer than just “mature”.
Reply: We rewrote this sentence in the revised manuscript to make a clear description.
Page 3-line L220-221in the revised version: “Two healthy mature leaves, the seventh and eighth leaves counting from the top, were utilized for infestation.”
L.127: the cycle of the pest was not explained before, needed for the 4th instar to be understood.
Reply: The larvae of H. cunea have six instars per generation. The larvae from the 1st to 3rd instars exhibit gregarious behavior. Starting from the 4th instar, the larvae begin to feed separately, and enter a period of voracious feeding. We added the information in the revised manuscript.
Page 3-line L222-223 in the revised version: “Fourth instar larvae of H. cunea which begun to feed separately and entered the voracious feeding phase were used for the infestation.”
L.129: how was the extent of leaf consumption measured? ImageJ?
Reply: The extent of leaf consumption was measured by Image J software. We added the information in the revised manuscript.
Page 3-line L226-227 in the revised version: “At 24 hpi, 48 hpi, and 96 hpi, the leaf consumption was measured by Image J software, and the weight of the larvae was recorded”
L.137-138: had to be rephrased. It’s confusing with the paragraph before “followed by storage at -80°C for subsequent metabolite analysis.”: was analyzed right after sampling or later?
Reply: We rewrote this sentence in the revised version.
Page 3-line L237-238 in the revised version: “The frozen leaves were taken out of the refrigerator and ground into a powder.”
L.140: is there an available publication for this kit?
Reply: We added the references in the revised manuscript.
Page 4-line L322-324 in the revised version: “The activities of POD and PPO enzymes in each cultivar were then determined using the POD and PPO Activity Assay Kit (Solarbio), with specific assay methods detailed in the kit's instructions [29, 30]”
[29] Wang, Y.; Yang, L.; Zhou, X.; Wang, Y.; Liang, Y.J.; Luo, B.S.; Dai, Y.H.; Wei, Z.L.; Li, S.L.; He, R.; Ding, W. Molecular mechanism of plant elicitor daphnetin-carboxymethyl chitosan nanoparticles against Ralstonia solanacearum by activating plant system resistance. Int. J. Biol. Macromol. 2023, 241, 124580. https://doi.org/10.1016/j.ijbiomac.2023.124580.
[30] Ge, X.L.; Zhang, L.; Du, J.J.; Wen, S.S.; Qu, G.Z.; Hu, J.J. Transcriptome analysis of Populus euphratica under salt treatment and PeERF1 gene enhances salt tolerance in transgenic Populus alba × Populus glandulosa. Int. J. Mol. Sci. 2022, 23, 3727. https://doi.org/10.3390/ijms23073727.
L.140-143: Either this needs to be rephrased or the protocol does not add up. On l.115-116, it was mentioned: “For each cultivar, 30 seedlings were chosen for the experiment, with 15 designated for infestation and the remaining 15 serving as non-infestation controls”, which rounds up to 60 seedlings in total.
However, in this paragraph, we have:
“for every cultivar” -> x2
“at each time point” -> x3
“both infestation and non-infestation samples” -> x2
“three plants, representing three biological replicates” -> x3
“with each biological replicate having three technical replicates” -> x3
Which rounds up to 108 seedlings. Please clarify this key point.
Reply: We are sorry for a wrong presentation. We rewrote the sentences to make a clear description for sampling. For each cultivar, 60 seedlings were chosen for the experiment, with 30 designated for infestation and the remaining 30 serving as non-infestation controls. After infestation, the samples were collected at three time points. For every cultivar at each time point, there were 10 seedlings inoculated with H. cunea, and 10 seedlings with controls. Among the 10 seedlings that were either inoculated or controls, 7 were used for metabolic profiling, and 3 were used for enzyme activity analysis.
Page 3-line L219-223 in the revised version: “For each cultivar, 60 seedlings were chosen for the experiment, with 30 designated for infestation and the remaining 30 serving as non-infestation controls…… For every cultivar at each time point, 10 seedlings were inoculated with H. cunea, and another 10 seedlings served as uninoculated controls. Among the 10 seedlings that were either inoculated or controls, 7 seedlings were used for metabolic profiling, and 3 ones were used for enzyme activity analysis.”
L.145-168: in this whole paragraph, clarify what has been modified from the protocols used in your references.
Reply: We revised this paragraph by removing the same method from the references and added a description for the modifications made to the method.
Page 4-line L328-329 in the revised version: “The frozen leaf tissues were ground into a fine powder using a mortar and pestle. A sample of approximately 100 mg of this powder was used for the extraction of metabolites according to previous methods with some modifications [27,31]. Add the ground powder to 1.5ml of the extraction solution (methanol: chloroform: water in a ratio of 5:2:2). To this mixture, internal quantitation standards: 5 µL of nonadecanoic acid at a concentration of 2.0 mg/mL and 80 µL of ribitol at a concentration of 0.2 mg/mL were added. After extraction, 500 µL of the polar extract and 200 µL of the lipophilic extract were evaporated to dryness under a stream of nitrogen in a vacuum rotary evaporator, without the application of heat. To the dried residues, 50 µL of methoxy amine hydrochloride (20 mg/mL, Sigma-Aldrich, St. Louis, MO, USA) in pyridine (ACS grade) was added, and the mixture was incubated for 2 hours at 30 ℃ to derivatize the metabolites. Following this, 100 µL of N-Methyl-N-trifluoroacetamide (MSTFA) was added to the derivatized extracts and incubated for an additional 30 minutes at 37 ℃ to complete the chemical reaction. The derivatized samples were then analyzed using a LECO Pegasus IV GC–TOF/MS system equipped with a DB-5 ms capillary column (0.25 mm, 30 m×0.32 mm) for the separation of the metabolites. The temperature program for the chromatographic separation of polar and non-polar compounds was based on the method previously described by Wang et al [27].”
L.171-172: little details on these two criteria.
Reply: We added details for the data analysis, and description for the two criteria.
Page 4-line L347-356 in the revised version: “The data underwent mean-centering and autoscaling to achieve a unit variance, followed by analysis using partial least squares regression - discriminant analysis (PLS-DA) with the SIMCA-P 12.0 software package (Umetrics, Umeå, Sweden). In the PLS-DA model's loadings plot, metabolites identified by a variable importance for the project (VIP) >1.0 were considered responsible for profile differences and were selected for further examination. Then, nonparametric Mann-Whitney U tests were conducted to compare the abundance of the above metabolites across different cultivars. Metabolites were deemed differential if they met two criteria: a VIP score greater than 1.0 and a Mann-Whitney P-value below 0.05.”
L.173-174: National Institute of Standards and Technology from where?
Reply: National Institute of Standards and Technology (NIST) is from USA. We added the country in the revised version.
Page 4-line L357-358 in the revised version: “These differential metabolites were identified by comparing the mass of fragments with the National Institute of Standards and Technology (NIST, USA) standard mass spectral databases using a similarity of 80%.”
L.177-178: Is Arabidopsis thaliana the most documented? Nothing closer phylogenetically speaking that has a well-documented metabolome?
Reply: In the web-site of Metabo-Analyst, there are 10 plant pathway libraries, but there is no database for poplar and its closely related species within the Populus genus. Among these databases, the library of Arabidopsis is the most comprehensive and the most documented one, so we chose it.
3.Results
L.181-186: all that should be removed, it’s Material and Methods, not results. Moreover, the uncertainty of the number of used saplings is also present at l.184.
Reply: We deleted the sentences related to Material and Methods, and rewrote this paragraph to make a clear presentation for the results. We also clarified the number of used saplings for each cultivar in the paragraph of 2.3. Treatment of Poplar Trees with H. cunea.
Page 4-line L368-445 in the revised version: “The seedlings were infested with larvae of H. cunea as shown in Figure 1A. post-infestation, the extent of leaf area consumed and the corresponding larval weight were documented to ascertain the comparative resistance levels of the two cultivars. After feeding for the same duration, the consumed leaf area of '2KEN8' was significantly smaller than that of 'Nankang'. Figure 1B shows the leaves of the two varieties at 48 hours after infestation. The diminished feeding area on the '2KEN8' leaves suggested a potential for reduced weight gain in larvae that feed on this cultivar compared to those feeding on 'Nankang'. Indeed, measurements of larval average weight indicated that although the larvae's weight was similar before infestation for both varieties, after 96 hours of feeding, there was a significant change in weight between larvae feeding the two varieties. The average weight of larvae feeding on 'Nankang' leaves was 220mg, with an increase of more than twice the body weight, while the larvae feeding on '2KEN8' leaves only had an average weight of 140mg, with an increase of only one fold in the body weight (Figure 1C). These phenotypic outcomes corroborated that ‘2KEN8’ was more resistant to H. cunea compared to ‘Nankang’.”
Page 3-line L219-223 in the revised version: “For each cultivar, 60 seedlings were chosen for the experiment, with 30 designated for infestation and the remaining 30 serving as non-infestation controls…… For every cultivar at each time point, 10 seedlings were inoculated with H. cunea, and another 10 seedlings served as uninoculated controls. Among the 10 seedlings that were either inoculated or controls, 7 seedlings were used for metabolic profiling, and 3 ones were used for enzyme activity analysis.”
L.187: notably less extensive leaf damage: just visually estimated or actually measured?
Reply: The extent of leaf consumption was measured by Image J software.
Page 3-line L226-227 in the revised version: “At 24 hpi, 48 hpi, and 96 hpi, the leaf consumption was measured by Image J software…”
L.191: were there any statistical tests corroboring this result? Just comparing 145mg with 220mg is not enough to justify the difference.
Replay: At each time point, the average weight of larvae feeding on each cultivar was derived from ten larvae sampled from ten inoculated seedlings. At each time point, the difference in average weight between larvae feeding on the two cultivars was compared using a Student's t-test. The statistical tests supporting this result were added in the methods.
Page 4-line L360-363 in the revised version: “Statistical analysis was performed by SPSS version 17.0 (SPSS Inc., Chicago, IL, USA). Student's t-test was used to compare the differences in average weight between larvae feeding on the two cultivars (P < 0.05).”
L.197-198: which test was applied to check the difference?
Replay: Student's t-test was used to check difference. ** represented the difference at a significant level at P<0.01.
Page 4-line L360-363 in the revised version: “Statistical analysis was performed by SPSS version 17.0 (SPSS Inc., Chicago, IL, USA). Student's t-test was used to compare the differences in average weight between larvae feeding on the two cultivars (P < 0.05).”
Page 5-ine L449-450 in the revised version: “The two stars represented the difference at a significant level at P<0.01.”
Figure 2: the patterns in the rectangle are not distinct enough to be recognized.
Replay: We redrew this figure to make a clear representation.
Page 6-line L536-539 in the revised version:
“Figure 2. Changes of POD and PPO activities in leaves of the infestationed group and the control group at 24 hpi and 48 hpi. (A) POD activity; (B) PPO activity. C: control group. T: infested group. Duncans' multiple range tests were performed to determine significant difference among inoculated and control samples. Different letters in the figure indicated significant differences (P<0.05).”
L.212: which test?
Replay: Duncans' multiple range tests were conducted to determine significant difference among inoculated and control samples (P < 0.05).
Page 6-line L537-539 in the revised version: “Figure 2. ….Duncans' multiple range tests were conducted to determine significant difference among inoculated and control samples. Different letters in the figure indicated significant differences (P<0.05).”
Figure 3: what is the colour code? What do the squares represent? What are the units of the axes? How were these profiles created? Indicate in the paragraph and in the legend of the figure!
Replay: Figure 3 is a PLS-DA score plot generated using SIMCA-P, designed to illustrate the metabolic profile differences among samples. In this plot, the same color represented 7 replicates of each material at the same time point in inoculated or control group. Each square represented an individual sample. The horizontal axis (X-axis) in the PLS-DA plot represented the first principal component (t1), which captured the direction of greatest variability in the data. The values on X-axis represent the projection of the samples onto this dimension, reflecting the primary source of variation. The vertical axis (Y-axis) represented the second principal component (t2), which accounts for the second greatest variability in the data, orthogonal to the first principal component. The values on Y-axis represented the projection of the samples onto this dimension, reflecting the secondary source of variation. We added the above information in the figure legend.
Page 4-line L348-350 in the revised version: “The data underwent mean-centering and autoscaling to achieve a unit variance, followed by analysis using partial least squares regression - discriminant analysis (PLS-DA) with the SIMCA-P 12.0 software package (Umetrics, Umeå, Sweden).”
Page 6-line L555-573 in the revised version:
“Figure 3. PLS-DA score plot of the infested and control samples at 24 hpi, 48 hpi and 96 hpi.
The ellipses represent the Hotelling T2 with 95% confidence. t[1] and t[2] represent the first and second principal component, respectively. Each square represented an individual sample. The squares with same color represented 7 replicates of each material at the same time point in inoculated or control group. The samples on the left side of the figure were the control and infested groups of the resistant ‘2KEN8’, while those on the right side were the control and inoculated samples of the susceptible ‘Nankang’. The solid lines represented the trajectories of the inoculated samples, while the dashed lines represented the trajectories of the control samples.”
L.231: Indicate that the comparison is not represented.
Reply: We are sorry that we do not grasp the meaning of this question.
L.233: here it indicates 45 differential compounds against 35 in figure 4A. Which one is correct?
Figure 4: consider changing the colours. The multiple use of the same colours makes it confusing.
Replay: We are sorry that we made a mistake, and 35 is right. We corrected it in the revised version. To present more clearly, we rearranged the colors for different groups and provided explanations for the samples represented by the colors in figure legend.
Page 7-line L576-578 in the revised version: “Compared to their respective non-infestation controls, at 24 hpi, feeding of H. cunea in-duced 80 differential metabolites in ‘2KEN8’ and 35 in ‘Nankang’, with 23 of these being common differential metabolites (Figure 4A).”
Page 7-line L586 in the revised version:
Figure 4. Number of differential metabolites between control and infested samples. (A-C) Comparison of the differential compounds induced by feeding of H. cunea in ‘2KEN8’and ‘Nankang’ at 24 hpi (A), 48 hpi (B) and 96 hpi (C), respectively. The light purple and pale-yellow circles represented the differential metabolites between the inoculated (T) and control (C) group for ‘2KEN8’and ‘Nankang’, respectively. (D-E) Number of differential metabolites induced by H. cunea at the three time points in ‘2KEN8’ (D) and ‘Nankang’ (E). The pink, pistachio, and sky-blue circles represented the number of differential metabolites between the inoculated and control samples at 24h, 48h, and 96h, respectively. T/C: Differential compounds between infested and control group.
L.246: there are no “lowercase letters in the figure”. Forgotten?
Replay: We are sorry we make a mistake. We deleted the sentence “Lowercase letters in the figure indicated significant differences (P<0.05).” in the figure legend.
Page 7-line L587-594 in the revised version: “Figure 4. Number of differential metabolites between control and infested samples. (A-C) Com-parison of the differential metabolites induced by feeding of H. cunea in ‘2KEN8’ and ‘Nankang’ at 24 hpi (A), 48 hpi (B) and 96 hpi (C), respectively. The light purple and pale-yellow circles represented the differential metabolites between the infested (T) and control (C) group for ‘2KEN8’ and ‘Nankang’, respectively. (D-E) Number of differential metabolites induced by H. cunea at the three time points in ‘2KEN8’ (D) and ‘Nankang’ (E). The pink, pistachio, and sky-blue circles represented the number of differential metabolites between the infested and control samples at 24h, 48h, and 96h, respectively. T/C: Differential metabolites between infested and control group.”
L.249-261: this whole paragraph needs to be re-written because of poor English, especially of “former” and “latter”. For instance, in “constitutive differences between the resistant ‘2KEN8’ and the susceptible ‘Nankang’. The former shows significantly lower levels in ‘2KEN8’”, “the former” refers to “the resistant ‘2KEN8’”.
Replay: We rewrote this paragraph to make a clear presentation.
Page 7-line L596-682 in the revised version: “Metabolomic analysis indicated that differential metabolites induced by H. cunea are significantly enriched in the following pathways, primarily including the phenylpro-panoid pathway, flavonoid biosynthesis, and biosynthesis of unsaturated fatty acids. Within the phenylpropanoid pathway, phenylalanine, cinnamic acid, coumaric acid and sinapic acid exhibited constitutive differences between the two cultivars. The content of phenylalanine in the resistant ‘2KEN8’ was significantly lower than that in susceptible ‘Nankang’. However, the contents of cinnamic acid, coumaric acid, and sinapic acid were higher in ‘2KEN8’ than in ‘Nankang’. These compounds were also induced by H. cunea infestation. After infestation,the level of phenylalanine was induced to decrease at all three time points. Contrary to the change in phenylalanine, the contents of cinnamic acid and coumaric acid were induced to increase at 48hpi and 96hpi, and that of sinapic acid was elevated at 24 hpi and 48 hpi. For cinnamic acid, coumaric acid and sinapic acid, the fold increase in the resistant ‘2KEN8’ was slightly higher than susceptible ‘Nankang’ following infestation (Figure 5).”
L.246: All three compounds were induced by H. cunea infestation…
Reply: We corrected the sentence in the revised version.
Page 8-line L677 in the revised version: “These compounds were also induced by H. cunea infestation.”
L.275-279: belongs in the discussion, not the results.
Reply: We removed the sentence to the discussion in the revised version.
Page 11-line L983-985 in the revised version: “The unsaturated fatty acid, α-linolenic acid, is a precursor for the synthesis of jasmonic acid (JA), which is a crucial signaling molecule in plant defense responses to insects [46-48].”
Figure 5: way too small, not agreeable to read.
Which test was applied to differentiate the groups?
Reply: We redrew this figure to make it agreeable to read. Duncan's multiple range tests were performed to determine significant difference among inoculated and control samples. Different letters in the figure indicated significant differences (P<0.05). We added this information in the figure legend.
Page 8-line L698-703 in the revised version: “Figure 5. Relative contents of differential metabolites in pathways of phenylpropanoid and flavonoid biosynthesis. R: resistant ‘2KEN8’. S: susceptible ‘Nankang’. C: control group. T: infested group. Duncan's multiple range tests were performed to determine significant difference among inoculated and control samples. Different letters in the figure indicated significant differences (P<0.05).”
Figure 5. Relative contents of differential metabolites in pathways of phenylpropanoid and flavonoid biosynthesis. R: resistant ‘2KEN8’. S: susceptible ‘Nankang’. C: control group. T: infestation infested group. Duncan's multiple range tests were performed to determine significant difference among inoculated and control samples. Different letters in the figure indicated significant differences (P<0.05).”
Figure 6: nicer size! What are the deltas? Which test was applied to differentiate the groups?
Reply: In the pathway for biosynthesis of unsaturated fatty acids, Δ9, Δ12, and Δ15 indicated the specific positions on the carbon chain where desaturation reactions occur. To avoid confusing readers, we removed these symbols. Duncan's multiple range tests were performed to determine significant difference among inoculated and control samples. We added this information in the figure legend.
Page 9-line L768-770 in the revised version: “Duncan's multiple range tests were performed to determine significant difference among infested and control samples. Different letters in the figure indicated significant differences (P<0.05).”
4.Discussion
L.319-320: extensive research […] which? General references to start with! The rest of the paragraph is nicely documented.
Reply: We added the references in the revised version.
Page 10-line L835-837 in the revised version: “Extensive research has established phenolic substances as a pivotal class of anti-herbivory compounds as reviewed by Lattanzino et al [14] and Mithöfer et al [16].”
[14] Lattanzino, V.; Lattanzino, V.M.T.; Cardinali, A. Role of phenolics in the resistance mechanisms of plants against fungal pathogens and insects. Phytochem. 2006, 661, 23-67.
[16] Mithöfer, A.; Boland, W. Plant defense against herbivores: chemical aspects. Annu Rev Plant Biol. 2012, 63, 431-50. https://doi.org/10.1146/annurev-arplant-042110-103854.
L.336-340: redundant, could be simpler.
Reply: We rewrote this sentence in the revised version.
Page 10-line L852-855 in the revised version: “Furthermore, feeding by H. cunea triggered an increase in the levels of these metabolites, with the resistant ‘2KEN8’ showing a significantly greater increase than the susceptible ‘Nankang’, underscoring the involvement of these phenolic compounds in induced resistance to H. cunea.”
L.341: 4.3. Flavonoid and JA Signalling Pathway Contribute to the Earlier Eefensive Responses in'2KEN8'
Reply: We corrected the sentence in the revised version.
Page 10-line L856-857 in the revised version: “4.4. Flavonoid and Unsaturated Fatty Acids Contribute to the Earlier Defensive Responses in ‘2KEN8’.”
L.358: and the less gain of larval body weight
Reply: We corrected the sentence in the revised version.
Page 11-line L975-976 in the revised version: “…. and the lesser gain of larval body weight observed with this cultivar….”
L.364: JA is a kind of plant hormones derived from unsaturated fatty acids
Reply: Thank you. We have rewritten this sentence and no longer use this sentence.
L.372: This study
Reply: We corrected the sentence in the revised version.
Page 10-line L827 in the revised version: “The present study found….”
5.Conclusion
The whole paragraph is redundant.
L.380: relatively slower/lesser weight gain
L.382: following the infestationed by H.cunea
L.389-391: which experiments could be performed?
It lacks a conclusive sentence.
Reply: We refined this paragraph to present a concise conclusion.
Page 11-line L996-1006 in the revised version: “The present study identified that larvae feeding on the '2KEN8' showed a significantly slower weight gain and reduced leaf consumption compared to those on the 'Nankang', thereby confirming that '2KEN8' possesses greater resistance to H. cunea than 'Nankang'. Analyze on the dynamic metabolic responses of the two varieties to the insect showed that the resistant ‘2KEN8’ initiated earlier and stronger defense responses, including higher increase on the activity of defensive enzymes and more accumulation of compounds such as phenolics, flavonoids, and unsaturated fatty acids. Higher levels of defensive enzymes and metabolites might restrict the feeding of larvae, thereby contributing to the resistance of '2KEN8' to H. cunea. The present study provides an investigation on chemical defense of trees to H. cunea, and lays a foundation for further elucidation of the molecular mechanisms underlying tree resistance to this insect.”

Reviewer 2 Report
Comments and Suggestions for Authors

Author Response
Dear editor:
Thank you very much for your attention and careful consideration to our manuscript. We appreciate you very much for your positive and constructive comments and suggestions on our manuscript.
We have carefully considered your suggestion and comments and revised our manuscript according to these precious comments. We have submitted revision manuscript using online system, and we also have uploaded a marked-up copy of the changes made from the previous manuscript as attachment by the E-mail.
The main corrections and the responds to the comments point to point are listed blow:
Responses to the comments and suggestions of Reviewer #2
Comments:
Poplar trees are significant for both economic and ecological purposes, and the fall webworm (Hyphantria cunea Drury) poses a major threat to their plantation in China. This study focuses on analyzing the defense and metabolic responses of resistant “2KEN8” and susceptible “Nankang” with or without feeding by H. cunea, which is of great significance for resistance breeding of Poplar forests. There are still some issues with the manuscript, and it is recommended to revise it according to the following suggestions before publishing. The main problems are as follows:
The quality of all Figures is unclear, update required.
Reply: We redrew the figures in the revised version.
Page 5-line L446 in the revised version:
Figure1 Figure 1. Evaluation of resistance to H. cunea in the two poplar clones. (A) Two newly mature leaves, the seventh and eighth leaves of the seedling were used for larval infestation; (B) Leaf consumption at 48 hpi; (C) Changes of larva average weight when feeding for 0h and 96h. The two stars indicated a significant level with a P-value less than 0.01.
Page 6-line L535 in the revised version:
Figure 2. Changes of POD and PPO activities in leaves of the infested and the control group at 24 hpi and 48 hpi. (A) POD activity; (B) PPO activity. C: control group. T: infested group. Duncan's multiple range tests were performed to determine significant difference among inoculated and control samples. Different letters in the figure indicated significant differences (P<0.05).
Page 6-line L554 in the revised version:
Figure 3. PLS-DA score plot of the infested and control samples at 24 hpi, 48 hpi and 96 hpi. The ellipses represented the Hotelling T2 with 95% confidence. t[1] and t[2] were the first and second principal component, respectively. Each square represented an individual sample. The squares with same color were 7 replicates of each material at the same time point in infested or control group. The samples on the left side of the figure were the control and infested groups of the resistant ‘2KEN8’, while those on the right side were the control and inoculated samples of the susceptible ‘Nankang’. The solid lines represented the trajectories of the inoculated samples, while the dashed lines represented the trajectories of the control samples.
Page 7-line L586 in the revised version:
Figure 4. Number of differential metabolites between control and infested samples. (A-C) Comparison of the differential metabolites induced by feeding of H. cunea in ‘2KEN8’ and ‘Nankang’ at 24 hpi (A), 48 hpi (B) and 96 hpi (C), respectively. The light purple and pale-yellow circles represented the differential metabolites between the infested (T) and control (C) group for ‘2KEN8’ and ‘Nankang’, respectively. (D-E) Number of differential metabolites induced by H. cunea at the three time points in ‘2KEN8’ (D) and ‘Nankang’ (E). The pink, pistachio, and sky-blue circles represented the number of differential metabolites between the infested and control samples at 24h, 48h, and 96h, respectively. T/C: Differential metabolites between infested and control group.
Page 8-line L698 in the revised version:
Figure 5. Relative contents of differential metabolites in pathways of phenylpropanoid and flavonoid biosynthesis. R: resistant ‘2KEN8’. S: susceptible ‘Nankang’. C: control group. T: infested group. Duncan's multiple range tests were performed to determine significant difference among inoculated and control samples. Different letters in the figure indicated significant differences (P<0.05).
Page 9-line L765 in the revised version:
Figure 6. Relative abundance of differential metabolites in the pathway for biosynthesis of unsaturated fatty acids. R: resistant ‘2KEN8’. S: susceptible ‘Nankang’. C: control group. T: infested group. Duncan's multiple range tests were performed to determine significant difference among infested and control samples. Different letters in the figure indicated significant differences (P<0.05).
Line119:What is the feeding method for larvae? Artificial feed or plant leaves? If it is feed, please explain the feed formula clearly.
Reply: The fourth instar larvae of H. cunea were used for inoculating poplar seedlings. From newly hatched to third instar larvae, they were fed with an artificial diet. The components and weight ratios of the artificial diet were as follows: wheat germ 7%, sucrose 4%, protein 5%, Weiser's salts 0.8%, sorbic acid 1%, methyl 4-hydroxybenzoate 0.4%, ascorbic acid 0.17%, vitamin B 0.12%, agar 1.4%, cholesterol 0.11%, water 80%. We added this information in the revised version.
Page 3-line L208-213 in the revised version: “After the larvae hatched, they were reared with an artificial diet in the plastic cups in insectarium (23±1℃; 65% relative humidity; 16h light: 8h dark cycle) until the fourth instar. The components and weight ratios of the artificial diet were as follows: wheat germ 7%, sucrose 4%, protein 5%, Weiser's salts 0.8%, sorbic acid 1%, methyl 4-hydroxybenzoate 0.4%, ascorbic acid 0.17%, vitamin B 0.12%, agar 1.4%, cholesterol 0.11%, water 80%.”
Line129:What is the basis for setting 24, 48, and 96? Why not 24, 48, 72,96?
Reply: Before conducting the present experiment, we performed a preliminary analysis on the control and treated groups of resistant cultivars ‘2KEN8’ at 12h, 24h, 48h, 72h, and 96h after infestation with H. cunea. The results showed that the metabolic profiles at 12h and 24h, as well as 48h and 72h after infestation, were relatively similar. Therefore, we chose three time points (24h, 48h, and 96h) with more pronounced changes in the metabolic profile, to analyze the difference between resistant and susceptible varieties.
Lines 181 to 186: The content from lines 181 to 186 should be adjusted to the Materials and Methods section, including Figure 1A
Reply: We deleted the sentences related to Material and Methods, and rewrote this paragraph to make a clear description for the results.
Page 4-line L368-445 in the revised version: “The seedlings were infested with larvae of H. cunea as shown in Figure 1A. post-infestation, the extent of leaf area consumed and the corresponding larval weight were documented to ascertain the comparative resistance levels of the two cultivars. After feeding for the same duration, the consumed leaf area of '2KEN8' was significantly smaller than that of 'Nankang'. Figure 1B shows the leaves of the two varieties at 48 hours after infestation. The diminished feeding area on the '2KEN8' leaves suggested a potential for reduced weight gain in larvae that feed on this cultivar compared to those feeding on 'Nankang' leaves. Indeed, measurements of larval weight indicated that although the larvae's weight was similar before infestation for both varieties, after 96 hours of feeding, there was a significant change in weight between the two varieties. The average weight of larvae feeding on 'Nankang' leaves was 220mg, with an increase of more than twice the body weight, while the larvae feeding on '2KEN8' leaves only had an average weight of 140mg, with an increase of only one-fold in the body weight (Figure 1C). These phenotypic outcomes corroborated that ‘2KEN8’ was more resistant to H. cunea compared to ‘Nankang’.”
Line210:The quality of Figure 2 is unclear.
Reply: We redrew the Figure 2 in the revised version.
Page 6-line L535 in the revised version: “.”
Figure 2. Changes of POD and PPO activities in leaves of the infested and the control group at 24 hpi and 48 hpi. (A) POD activity; (B) PPO activity. C: control group. T: infested group. Duncan's multiple range tests were performed to determine significant difference among inoculated and control samples. Different letters in the figure indicated significant differences (P<0.05).
Suggestions for modifying the discussion section the discussion section suggests adding content on insect feeding induced plant resistance.
Reply: We added a paragraph to discuss the content on insect feeding induced resistance and the involved metabolites.
Page 10-line L816-831 in the revised version:
“4.2. Feeding of H. cunea triggered stronger metabolic responses in the Resistant ‘2KEN8’
Induced resistance is a cost-effective and efficacious strategy commonly employed by plants to defend against insect herbivory. Metabolites play a significant role in insect-feeding-induced resistance responses [16]. Research on various plant-insect interactions has shown that the common metabolites involved in the induced responses mainly include phenolic acids, tannins, flavonoids, alkaloids, terpenoids, steroids, fatty acid derivatives, and glycosides [16-18]. These defensive metabolites are often species-specific. In poplar trees, hydrolysable tannins have been reported to participate in induced resistance against Lymantria dispar, phenolic glycosides in resistance responses to Phyllocnistis populiella, and phenolic acids and phenolic glycosides in resistance reactions to Apriona germari (Hope) [22,25,26]. It is likely that even within the same species, the resistance compounds elicited in response to different insects' feeding may differ. The present study found that feeding by H. cunea induced accumulations of phenolic acids, flavonoids, and unsaturated fatty acids in poplar leaf tissues. In the resistant ‘2KEN8’, the accumulations occured earlier and were more substantial suggesting that the three kinds of metabolites are involved in the poplar's induced resistance response to H. cunea.”
Line341: The research content did not involve JA. Why discuss JA Signalling Pathway in 4.3? Suggest changing 4.3 to discuss differential metabolites induced by insect feeding in plants.
Reply: Thank you for your suggestion. We revised the title of 4.3. section in the discussion, and rewrote the secondary paragraph. We mentioned JA because we found that the precursor of JA synthesis, α-linolenic acid, as well as its upstream substances linoleic acid and oleic acid, were significantly induced to increase after inoculation. Therefore, we speculated that the JA signaling pathway was involved in the induced resistance of poplar to H. cunea. However, as you pointed out, we did not measure the content of JA, so this was just a speculation, and thus we reduced the description about JA signaling pathway in this section.
Page 11-line L978-994 in the revised version: “In the present study, levels of several metabolites in the pathway for biosynthesis of unsaturated fatty acids, including α-linolenic acid, linoleic acid and oleic acid, were significantly increased after feeding by H. cunea, and enhanced more in the resistant ‘2KEN8’. These findings were consistent with the previous studies which identified that increase in levels of unsaturated fatty acids were associated with the resistance to insects [27,45]. The unsaturated fatty acid, α-linolenic acid, is a precursor for the synthesis of jasmonic acid (JA), which is a crucial signaling molecule in plant defense responses to insects [46-48]. Change on levels of α-linolenic acid indicated that the JA-mediated defense response was initiated earlier in resistant ‘2KEN8’ than in susceptible ‘Nankang’. Additionally, some studies have shown that the signaling molecule JA could regulate the accumulation of flavonoids. Exogenous treatment with methyl jasmonate (MeJA) could enhance the content of flavonoids in plants, thereby improving their resistance to pests [49-52]. So, it was likely that feeding by H. cunea induced an intensification of unsaturated fatty acid synthesis, leading to a rapid accumulation of the JA precursors. This could result in the accumulation of JA and the activation of the JA signaling pathway, thereby promoting the early accumulation of flavonoids and contributing to the inhibition of feeding and development of H. cunea larvae.”

Round 2
Reviewer 2 Report
Comments and Suggestions for Authors
The manuscript has been carefully revised and I agree to publish it.